# *Is Your Code Generated by ChatGPT Really Correct?* Rigorous Evaluation of Large Language Models for Code Generation

**Jiawei Liu**[I]*     **Chunqiu Steven Xia**[I]*     **Yuyao Wang**[U]     **Lingming Zhang**[I]

University of Illinois Urbana-Champaign[I]     Nanjing University[U]

`{jiawei6, chunqiu2, lingming}@illinois.edu`    `yuyao6@outlook.com`

## Abstract

Program synthesis has been long studied with recent approaches focused on directly using the power of Large Language Models (LLMs) to generate code. Programming benchmarks, with curated synthesis problems and test-cases, are used to measure the performance of various LLMs on code synthesis. However, these test-cases can be limited in both quantity and quality for fully assessing the functional correctness of the generated code. Such limitation in the existing benchmarks begs the following question: *In the era of LLMs, is the code generated really correct?* To answer this, we propose EvalPlus – a code synthesis evaluation framework to rigorously benchmark the functional correctness of LLM-synthesized code. EvalPlus augments a given evaluation dataset with large amounts of test-cases newly produced by an automatic test input generator, powered by both LLM- and mutation-based strategies. While EvalPlus is general, we extend the test-cases of the popular HUMANEVAL benchmark by $80\times$ to build HUMANEVAL$^+$. Our extensive evaluation across 26 popular LLMs (*e.g.,* GPT-4 and ChatGPT) demonstrates that HUMANEVAL$^+$ is able to catch significant amounts of previously undetected wrong code synthesized by LLMs, reducing the pass@$k$ by up-to 19.3-28.9%. We also surprisingly found that test insufficiency can lead to mis-ranking. For example, both WizardCoder-CodeLlama and Phind-CodeLlama now outperform ChatGPT on HUMANEVAL$^+$, while none of them could on HUMANEVAL. Our work not only indicates that prior popular code synthesis evaluation results do not accurately reflect the true performance of LLMs for code synthesis, but also opens up a new direction to improve such programming benchmarks through automated testing. We have open-sourced our tools, enhanced datasets as well as all LLM-generated code at https://github.com/evalplus/evalplus to facilitate and accelerate future LLM-for-code research.

## 1 Introduction

Automatically generating programs that accurately correspond to user intents is a long-standing challenge in computer science known as program synthesis [21]. In the past few decades, classical program synthesis techniques have been developed, including deductive synthesis [19, 39, 62], inductive synthesis [20, 58] and neural-guided synthesis [29]. More recently, with the advent of Large Language Models [61, 6] (LLMs) and the abundance of open codebase, researchers have been focusing on applying LLMs for direct code generation. LLMs like CODEX [11] and CodeGen [46]

---

*Equal contribution. Author ordering is decided by *Nigiri*.

37th Conference on Neural Information Processing Systems (NeurIPS 2023).

```
[4,3,2,8], []          def common(l1: list, l2: list):                              []
[5,3,2,8], [3,2]           """Return sorted unique common elements for two lists"""  [2,3]
[4,3,2,8], [3,2,4]         common_elements = list(set(l1).intersection(set(l2)))    [2,3,4]
HUMANEVAL inputs           common_elements.sort()                                   ✓ correct
                           return list(set(common_elements))
[6,8,1], [6,8,1]                                                                     [8,1,6]
HUMANEVAL+ input                     ChatGPT synthesized code                       ✗ not sorted!
```

Figure 1: Exemplary wrong code synthesized by ChatGPT for HUMANEVAL #58.

perform code generation by autoregressively predicting the next token given previous context, in the form of function signature and docstring that denote the desired program functionality. The generated code snippet is then combined with the context to form a complete function that aligns with the user intent. Leveraging both natural language understanding and generative power, LLMs have demonstrated impressive performance in code synthesis [3, 11].

The primary concern when it comes to LLM-generated code is correctness. Because two dramatically different code snippets can be semantically equivalent, classic NLP metrics like BLEU score [50] are no longer reliable in the context of program synthesis. Ideally, we would like to formally verify the correctness of LLM-provided solutions for any input, but verifying domain-specific problems through methods such as translation validation [36, 44, 4] is already challenging enough, let alone building a general verifier with absolute certainty to prove arbitrary problems, including those in code benchmarks. As such, existing code benchmarks (*e.g.,* HUMANEVAL [11]) heavily rely on manually constructed test-cases to evaluate LLM solutions. However, these tests often fall short in capturing all possible scenarios, as crafting high-quality tests is laborious. Consequently, we argue that current programming benchmarks are inadequate for assessing the actual correctness of LLM-generated code, leading to false confidence in the results. Specifically, we have identified the following common limitations in existing LLM-for-code benchmarks:

- **Insufficient testing.** Current programming benchmarks often only include on average less than 10 tests for each coding problem. Furthermore, these tests are relatively too simple to fully explore the functionality of the code or corner cases. Figure 1 shows an incorrect code sample synthesized by ChatGPT [48] to return the sorted unique common elements from two lists. At first glance, the function looks correct and computes the desired output when using the base test inputs from HUMANEVAL. However, in the return statement, it incorrectly converts the intermediate list to a set which no longer preserves the order of the sorted list. This example shows that a logically flawed solution can still pass all simple tests and be misconsidered as correct due to testing inadequacy.
- **Imprecise problem description.** The input for code generation includes natural language descriptions in addition to the function signature. These task descriptions in existing benchmarks are oftentimes too vague to fully clarify the expected program behaviors. For example, the input docstring may not specify the expected input domain (*e.g.,* only positive integers) or how the function should handle exceptions. As a result, such programming problems can be interpreted differently by LLMs against the actual tests, leading to *capable* LLMs misjudged as incapable.

These limitations are common across many popular code generation benchmarks [11, 3, 33]. This not only questions the validity of the impressive performance claimed by prior work but also sets a challenge on how to properly evaluate the LLM coders. In this paper, we aim to address this fundamental evaluation challenge and ask the introspective question: *Is the code generated by LLMs really correct?*

**Our proposal.** In this work, we set out to answer the important question and *evaluate* the evaluation dataset. Consequently, we build EvalPlus – an evaluation framework to improve existing code benchmarks in order to precisely evaluate the functional correctness of LLM-generated code. At the heart of EvalPlus is an automatic test input generation engine which augments existing code benchmarks by generating interesting test inputs to fully exercise the code solution and check its functional correctness by cross-checking the ground-truth implementation. Specifically, EvalPlus adopts both LLM- and mutation-based [57, 74, 47] methods to automatically generate and diversify additional test inputs. EvalPlus first uses ChatGPT [48] to generate a set of high-quality seed inputs that aim to test difficult corner cases and functionalities of the program within the valid input structure. Using these high-quality seed inputs, EvalPlus then performs type-aware mutation to efficiently generate a large number of additional test inputs. These newly generated test inputs are then used to evaluate the LLM-generated code through differential testing [40] against the ground-truth implementation. Furthermore, as an option to speed up evaluation, EvalPlus also builds minimal test-suites by only

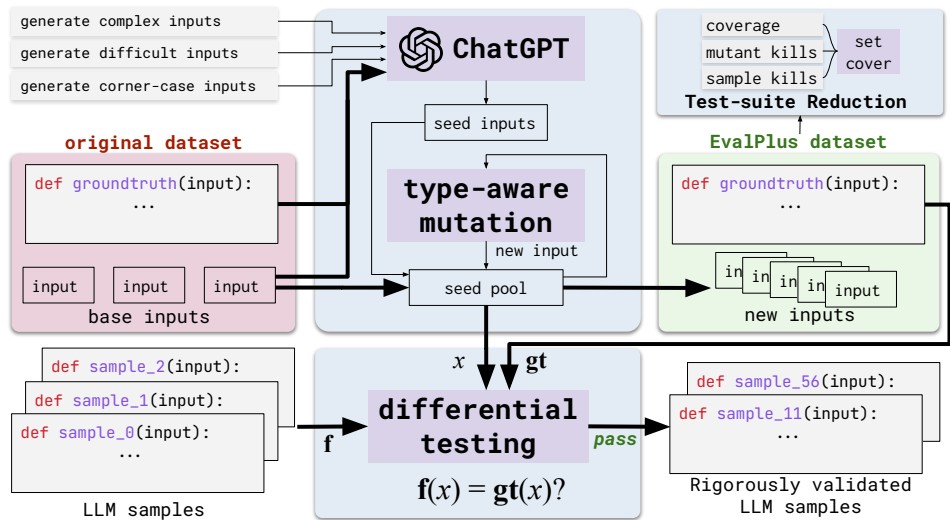

Figure 2: Overview of EvalPlus

including the most valuable test-cases, which are selected by running a greedy set cover algorithm to preserve the same code coverage [24], mutation analysis [7] as well as empirical LLM sample killings.

**Contribution.** Our work revisited and proposed to automatically improve code benchmarks for LLMs:

- **Study:** We are the first to study the test inadequacy problem in current programming benchmarks which can lead to largely over-approximated functional correctness. Our study also opens up a new research direction for precisely and rigorously evaluating LLM-synthesized code.
- **Approach:** We propose EvalPlus – an evaluation framework to reveal the real correctness of LLM-synthesized code. The test-case generation approach of EvalPlus combines the emerging LLM-based and traditional mutation-based test input generation. It first uses LLM-based strategy to bootstrap the test generator with high-quality seed inputs and then further extends large amounts of inputs via type-aware mutation. We then optionally "distill" the generated tests to a much smaller yet almost equivalently effective test-suite via greedy set covering. We also propose to annotate each programming tasks using program contracts to filter out invalid inputs.
- **Results:** EvalPlus extends the popular HUMANEVAL benchmark to create HUMANEVAL+, improving the test-case scale by $80\times$. Through test-suite reduction, we also produce HUMANEVAL+-MINI which distills HUMANEVAL+ tests by $47\times$ while still achieving a similar level of testing effectiveness. Our extensive evaluation over 26 popular LLMs surprisingly finds that the pass@$k$ on the new dataset is up-to 19.3-28.9% (for different $k$s) lower than the base HUMANEVAL, showing that testing insufficiency can largely affect the result analysis for almost all recent work on LLM-based code generation. Meanwhile, on the original HUMANEVAL both of the 34B WizardCoder-CodeLlama [38] and Phind-CodeLlama [52] models are deemed to be no better than ChatGPT, while HUMANEVAL+ corrected the ranking and shows that the two open-source models are actually better. Additionally, we even found that the ground-truth solutions of HUMANEVAL can be erroneous, further calling into question the quality of code synthesis benchmarks.

## 2 Approach

Figure 2 shows the overview of EvalPlus. We first take in as input the original dataset containing the ground-truth implementation as well as the base test inputs. EvalPlus starts with constructing a prompt using the original ground-truth, exemplary test inputs as demonstration, and a specialized instruction to query ChatGPT and generate a set of high-quality seed inputs. ChatGPT, by following base input formats and inspecting the ground-truth solution, can serve as a vehicle to generate valid yet rigorous test inputs. Starting from these seed inputs, we then perform type-aware mutation to quickly generate numerous new inputs together with seed inputs to extensively evaluate the functional correctness of LLM-generated code. We use differential testing [40] as the oracle to cross-check the output of the ground-truth and LLM-generated solution. As an option to speed up evaluation, EvalPlus

Table 1: List of basic type-aware mutations over input $x$.

| Type | Mutation | | Type | Mutation |
|------|----------|---|------|----------|
| `int`\|`float` | Returns $x \pm 1$ | | `List` | $\left\{\begin{array}{l}\text{Remove/repeat a random item } x[i] \\ \text{Insert/replace } x[i] \text{ with } \texttt{Mutate}(x[i])\end{array}\right.$ |
| `bool` | Returns a random boolean | | `Tuple` | Returns `Tuple(Mutate(List(`$x$`)))` |
| `NoneType` | Returns `None` | | `Set` | Returns `Set(Mutate(List(`$x$`)))` |
| `str` | $\left\{\begin{array}{l}\text{Remove a sub-string } s \\ \text{Repeat a sub-string } s \\ \text{Replace } s \text{ with } \texttt{Mutate}(s)\end{array}\right.$ | | `Dict` | $\left\{\begin{array}{l}\text{Remove a key-value pair } k \to v \\ \text{Update } k \to v \text{ to } k \to \texttt{Mutate}(v) \\ \text{Insert } \texttt{Mutate}(k) \to \texttt{Mutate}(v)\end{array}\right.$ |

runs set covering to minimize the generated test-suite while preserving the same level of testing effectiveness. As the final output, EvalPlus obtains a augmented benchmark using the generated high-quality test inputs to fully evaluate the functional correctness of LLM-synthesized code.

## 2.1 Automated Test Input Generation

**Seed initialization via ChatGPT.** EvalPlus first uses ChatGPT to generate a set of high-quality seed inputs for later mutation. Following Figure 2, we construct a prompt using *(i)* the ground-truth solution of the problem for ChatGPT to inspect; *(ii)* a set of test inputs as demonstration; and *(iii)* an instruction to encourage ChatGPT to come up with interesting inputs. Specifically, each prompt starts with the ground-truth implementation and then randomly sampled test inputs from the existing dataset. We then finalize the prompt with a selected instruction in Figure 2 and query ChatGPT to produce new inputs. EvalPlus aims to leverage the powerful understanding ability of ChatGPT to learn both the valid input formats (*e.g.,* variable types) as well as the desired functionality of the ground-truth solution in order to produce meaningful test inputs to reveal bugs in incorrectly synthesized code. Programs can have their own expected input formats, where invalid inputs should not be passed into the function as they can incur undefined behaviors to create false-positives in differential testing. As such, we filter out any invalid inputs which violate the input precondition required by the ground-truth implementation.

By using ChatGPT as an automated generation engine, we can generate inputs that are valid even under semantic constraints. For example, a programming problem may require the input to conform to a specific structure (*e.g.,* a palindrome). Such semantic constraints can be extremely difficult for traditional input generators to satisfy. However, ChatGPT is unsuitable for large amounts of automated test generation due to undesired speed and cost of querying such a large model. To address this, we perform type-aware input mutation starting from high-quality seed inputs generated by ChatGPT.

**Type-aware input mutation.** We follow a typical mutation-based fuzzing workflow [74, 57] to continuously create inputs: *(i)* a corpus of seed inputs from ChatGPT are used to initialize the seed pool and bootstrap the generation pipeline; *(ii)* each time an input (*i.e.,* seed) from the seed pool is randomly selected to be mutated to a new input (*i.e.,* mutant); and *(iii)* new inputs that comply with the program contract (§2.3) are added to the seed pool and we start over from *(ii)* to continue the generation process.

To efficiently create more valid inputs, we leverage type-aware mutation [66] in step *(ii)* which inspects the data types of the incoming valid seeds and generates new inputs that are structurally similar to the seeds. In Table 1 we illustrate the basic mutations used for different types of inputs. For simple primitive types such as `int` and `float`, the mutation is as simple as incrementing/decrementing the value. For compound types and the string type (*i.e.,* `str`), besides generally removing or repeating existing elements (or sub-strings for `str`), the elements and sub-strings can be mutated recursively according to their inner types. Such sub-mutants can then be used to replace existing items or add new items in a finer-grain manner. In addition, to alleviate generating inputs that violate subtle semantic constraints, following [23, 34], we additionally apply an ingredient mechanism to collect appeared data fragments and reuse them during mutation. In short, type-aware input mutation builds on the high-quality seed inputs produced by ChatGPT to generate large amounts of test inputs which we use as the final set of extensive test inputs to evaluate LLM-synthesized code.

## 2.2 Test-Suite Reduction

While the large number of newly generated tests in EvalPlus are effective in detecting incorrect code, the test execution can be costly. As an option to more efficiently evaluate LLM-generated code, we further investigate test-suite reduction strategies [75, 59], which aim to select a subset of the original test-suite while still maintaining the original test effectiveness. To perform test reduction, it is typically assumed that each test can fulfill a set of *testing requirements*. The problem can then be formalized as reducing the original test-suite $\mathcal{T}$ into $\mathcal{T}_{red}$, such that $\forall r \in \mathcal{R}$ ($\exists t \in \mathcal{T}$, $t$ satisfies $r \implies \exists t' \in \mathcal{T}_{red}, t'$ satisfies $r$). In other words, any testing requirement $r$ satisfied by the original test-suite should still be satisfied by the reduced one. Finding such minimal representative subset for a given test-suite is equivalent to the set covering problem [17]. To solve this problem effectively, it is crucial to define the testing requirements accurately. In this paper, we focus on the following types of requirements:

**Code coverage**: Code coverage [24] measures the amount of code elements (*e.g.,* statements or branches) executed by each test, and has been widely used in practice to measure test effectiveness. In this strategy, following traditional test-suite reduction [53] we leverage the widely used branch coverage as the testing requirement. In other words, the goal of using this metric is to only preserve a minimal subset of tests which can cover the same set of branches as the full tests.

**Mutant killings**: Coverage measures the extent to which the code has been executed; however, a high-coverage test-case is not necessarily effective in finding critical defects in its covered code. Consequently, researchers have proposed *mutation testing* [7] (also known as *mutation analysis*) to more precisely evaluate test effectiveness. In short, mutation testing applies a set of predefined *mutation rules* (*e.g.,* changing "<" and "≤") to the program under test (*i.e.,* the ground-truth solutions for this case) to create a large number of artificial buggy programs, each of which is called as a *mutant* and includes exactly *one* subtle bug seeded. In this way, the ratio of mutation bugs detected by the tests (also called *killed*) can be used to assess the test effectiveness. In fact, studies have shown that mutation testing can largely outperform code coverage in test effectiveness evaluation [51]. Following prior work [59], we also leverage the set of mutants killed by each test as our testing requirement. Consequently, the goal is to minimize the number of tests while still being able to detect the same set of mutation bugs.

**LLM sample killings**: Different LLMs could fail commonly over certain test-cases. Consequently, besides these theoretical metrics, we also use as a testing requirement by empirically looking at *sample killings*, *i.e.,* the set of wrong LLM samples that a test-case can detect and falsify. Of course, for a new LLM under evaluation, we do not have any test execution results for its code samples. Therefore, we only use the execution results for samples generated by other LLMs to evaluate test effectiveness for reduction (*i.e.,* leave-one-out cross validation [22]). As such, we minimize the number of tests while making sure that all incorrect samples synthesized by other models can be detected by the reduced test-suite.

Besides the above three strategies, we also investigate another strategy that merges all three testing requirements for reduction. That is, the goal is to minimize the number of tests while still maintaining the same branch coverage, mutant killing, and incorrect sample detection results.

## 2.3 Program Input Contracts

The goal of evaluating code synthesis is to check whether the synthesized code accurately reflects the desired user intent. This is done by using several test inputs and comparing the output of the generated code against that of the ground-truth solution. The prior sections demonstrated how to improve the test inputs used to more rigorously evaluate the synthesized code. However, these user intents (expressed as natural language docstring) can be too vague for LLMs to follow. As such, LLMs might allow for different interpretations of the desired functionality, input formats as well as how to handle corner cases.

To this end, we adopt a programming by contract [41] philosophy by systematically annotating function pre-conditions in form of code assertions (*e.g.,* `assert n > 0`), to ensure the test inputs for the function are well-formed. The benefits of the contracts are two-fold: *(i)* they can complement the automatic input generation steps to filter out any generated invalid inputs that violate the contracts. Such ill-formed inputs can incur undefined behaviors which are unreasonable to use for evaluating LLM-synthesized code; and *(ii)* they can serve as orthogonal descriptors together with the natural language description in the prompt for further clarification.

Table 2: Overview of EvalPlus-improved benchmarks.

| | #Tests | | | | #Tasks |
|---|---|---|---|---|---|
| | Avg. | Medium | Min. | Max. | |
| HUMANEVAL | 9.6 | 7.0 | 1 | 105[2] | |
| HUMANEVAL$^+$ | 764.1 | 982.5 | 12 | 1,100 | 164 |
| HUMANEVAL$^+$-MINI | 16.1 | 13.0 | 5 | 110 | |

## 3 Evaluation

**Setup.** Our evaluation focuses on using the unbiased version of pass@$k$ [11] to accurately assess the functional correctness of LLM-synthesized code. For generalizability, we conducted a comprehensive evaluation over 26 popular and state-of-the-art LLMs and a wide range of temperature settings. Specifically, following prior work [11, 46], for each model we perform: *(i)* random sampling to generate 200 program samples for each of the four temperature settings ($\{0.2, 0.4, 0.6, 0.8\}$); and *(ii)* greedy-search decoding. For random sampling, we show the best-performing pass@$k$ for each $k \in \{1, 10, 100\}$ and its corresponding temperature denoted by $T_k^*$. For greedy decoding, we only synthesize one deterministic sample for each task and evaluate its pass rate as pass@1$^\star$. By default we evaluate models under both setting *(i)* and *(ii)*, except for the two commercial models due to time and cost constraints: GPT-4 is only evaluated under greedy decoding, and ChatGPT is additionally evaluated on $0.8$-temperature random sampling.

While EvalPlus is general, this paper focuses on evaluating its effectiveness on HUMANEVAL [11], one of the most widely-used datasets for code generation[3]. HUMANEVAL consists of 164 human-written programming tasks, each of which provides a Python function signature and a docstring as the input to the LLM. Based on the input, LLMs complete a solution whose functional correctness is judged by a handful of manual test-cases (the first row in Table 2). As such, EvalPlus transforms HUMANEVAL to HUMANEVAL$^+$ by adding $80\times$ unique test-cases and fixing incorrect ground-truth solutions in HUMANEVAL. Specifically, for each task, based on around 30 ChatGPT-generated seed inputs which are produced using 3 separate prompts, we run type-aware mutation to generate 1000 additional inputs using one-hour budget. In HUMANEVAL$^+$, 83 out of the 164 programming tasks are annotated with hand-crafted contracts. Because EvalPlus requires ground-truth solutions to cross-check LLM-generated code, it is crucial to ensure the correctness of the ground-truths. However, by inspecting ground-truths in the original HUMANEVAL, we found over **10%** of them are incorrectly implemented. Therefore, as another contribution we carefully re-implemented and tested all ground-truths for HUMANEVAL$^+$. As an option to speed up evaluation, we build HU-MANEVAL$^+$-MINI which is minimized from HUMANEVAL$^+$ (smaller by $47\times$) yet preserves similar test effectiveness on the studied models. Lastly, more experimental setups are detailed in Appendix.

**Evaluation of LLMs.** Table 3 shows the pass@$k$ when evaluating LLMs using both the base HUMANEVAL and HUMANEVAL$^+$. We first observe that across all LLMs, models sizes and $k$ values, using HUMANEVAL$^+$, almost all pass@$k$ results **consistently drop** compared to using the base HUMANEVAL. Notably, the performance drop is significant with up-to 23.1% (pass@1$^\star$) / 19.3% (pass@1) / 24.9% (pass@10) / 28.9% (pass@100) reduction over the evaluated models. Such performance decrease is not only seen in popular open-source LLMs, such as the widely used CodeGen-16B [46] (18.5% reduction) as well as the emerging CODELLAMA-34B [54] (17.6%) and StarCoder [13] (14.1% reduction), but also observed in state-of-the-art commercial ChatGPT (12.6% reduction) and GPT-4 (13.1% reduction) models. Overall, our results overall confirm our hypothesis that the prior evaluation on HUMANEVAL is not robust enough to detect wrong code synthesized by LLMs. Not only are these LLMs widely used for daily programming but they also serve as common reference points for evaluating new code synthesis techniques. As such, evaluating on a more robust benchmark such as HUMANEVAL$^+$ is highly recommended in order to draw precise conclusions.

---

[2]There are four HUMANEVAL tasks (*e.g.,* `add(x, y)`) with over 100 "tests" (*i.e.,* implemented by cross-checking the ground-truth over random inputs). Without such, the maximum/average number is 26/7.3.

[3]Top-1 HuggingFace downloads on April, 2023. `https://hf.co/datasets?other=code-generation`

[4]To date, CodeGen2-16B is released with an unfinished checkpoint [45]. Nonetheless, we show its pass@1$^\star$.

Table 3: Evaluating LLMs on HUMANEVAL and HUMANEVAL⁺. All models, except for INCODER, CodeGen2, StarCoder and SantaCoder which perform infilling, use auto-regressive generation. $k=1^\star$ marks pass@1 done with greedy decoding. $T_k^*$ denotes the optimal pass@$k$ temperature.

| | Size | pass@k | $k=1^\star$ | $k=1$ | $k=10$ | $k=100$ | $T_1^*$ | $T_{10}^*$ | $T_{100}^*$ |
|---|---|---|---|---|---|---|---|---|---|
| GPT-4 [49] | N/A | base | 88.4 | | | | | | |
| | | +extra | 76.2 | | | | | | |
| Phind-CodeLlama [52] | 34B | base | 71.3 | 71.6 | 90.5 | 96.2 | .2 | .8 | .8 |
| | | +extra | 67.1 | 67.0 | 85.0 | 92.5 | .2 | .8 | .8 |
| WizardCoder-CodeLlama [38] | 34B | base | 73.2 | 61.6 | 85.2 | 94.5 | .2 | .8 | .8 |
| | | +extra | 64.6 | 54.5 | 78.6 | 88.9 | .2 | .8 | .8 |
| ChatGPT [48] | N/A | base | 73.2 | 69.4 | 88.6 | 94.0 | | | |
| | | +extra | 63.4 | 62.5 | 82.1 | 91.1 | | | |
| CODELLAMA [54] | 34B | base | 51.8 | 52.0 | 82.4 | 95.0 | .2 | .8 | .8 |
| | | +extra | 42.7 | 43.1 | 73.7 | 89.4 | .2 | .8 | .8 |
| | 13B | base | 42.7 | 44.6 | 77.6 | 92.7 | .4 | .8 | .8 |
| | | +extra | 36.6 | 37.4 | 69.4 | 88.2 | .4 | .8 | .8 |
| | 7B | base | 37.8 | 39.2 | 69.1 | 89.7 | .2 | .8 | .8 |
| | | +extra | 34.1 | 34.5 | 61.4 | 82.9 | .2 | .8 | .8 |
| StarCoder [13] | 15B | base | 34.1 | 32.2 | 56.7 | 84.2 | .2 | .8 | .8 |
| | | +extra | 29.3 | 27.8 | 50.3 | 75.4 | .2 | .8 | .8 |
| CodeGen [46] | 16B | base | 32.9 | 32.2 | 56.0 | 81.5 | .2 | .6 | .8 |
| | | +extra | 26.8 | 27.2 | 48.4 | 71.4 | .2 | .6 | .8 |
| | 6B | base | 29.3 | 27.7 | 46.9 | 72.7 | .2 | .6 | .8 |
| | | +extra | 25.6 | 23.6 | 41.0 | 64.6 | .2 | .6 | .8 |
| | 2B | base | 24.4 | 18.4 | 39.8 | 66.8 | .2 | .8 | .8 |
| | | +extra | 20.7 | 15.1 | 34.8 | 55.8 | .2 | .2 | .8 |
| CODET5+ [64] | 16B | base | 31.7 | 32.2 | 58.5 | 83.5 | .2 | .6 | .8 |
| | | +extra | 26.2 | 27.4 | 51.1 | 76.4 | .2 | .6 | .8 |
| MISTRAL [26] | 7B | base | 28.7 | 28.1 | 55.2 | 83.8 | .2 | .8 | .8 |
| | | +extra | 23.8 | 23.7 | 48.5 | 76.4 | .2 | .8 | .8 |
| CodeGen2 [45] | 16B[4] | base | 19.5 | | | | | | |
| | | +extra | 16.5 | | | | | | |
| | 7B | base | 18.3 | 17.9 | 30.9 | 50.9 | .2 | .6 | .8 |
| | | +extra | 16.5 | 15.9 | 27.1 | 45.4 | .2 | .6 | .8 |
| | 3B | base | 15.9 | 15.2 | 23.9 | 38.6 | .2 | .4 | .8 |
| | | +extra | 12.8 | 12.9 | 21.2 | 34.3 | .2 | .4 | .8 |
| | 1B | base | 11.0 | 10.2 | 15.1 | 24.7 | .2 | .6 | .6 |
| | | +extra | 9.1 | 8.7 | 13.7 | 21.2 | .2 | .6 | .6 |
| VICUNA [12] | 13B | base | 16.5 | 15.3 | 30.1 | 54.8 | .2 | .8 | .8 |
| | | +extra | 15.2 | 13.9 | 25.8 | 46.7 | .2 | .8 | .8 |
| | 7B | base | 11.6 | 10.9 | 23.8 | 42.3 | .2 | .6 | .6 |
| | | +extra | 11.0 | 10.3 | 20.3 | 35.0 | .2 | .6 | .6 |
| SantaCoder [2] | 1.1B | base | 14.6 | 16.6 | 29.2 | 45.4 | .4 | .6 | .8 |
| | | +extra | 12.8 | 14.2 | 26.2 | 40.6 | .4 | .6 | .8 |
| INCODER [18] | 6.7B | base | 15.9 | 15.6 | 27.7 | 45.0 | .2 | .4 | .6 |
| | | +extra | 12.2 | 12.4 | 22.2 | 38.9 | .2 | .6 | .6 |
| | 1.3B | base | 12.2 | 10.0 | 15.9 | 25.2 | .2 | .6 | .6 |
| | | +extra | 10.4 | 7.9 | 13.5 | 20.7 | .2 | .6 | .4 |
| GPT-J [63] | 6B | base | 12.2 | 11.3 | 17.7 | 31.8 | .2 | .6 | .6 |
| | | +extra | 10.4 | 9.5 | 15.2 | 25.9 | .2 | .6 | .6 |
| GPT-NEO [5] | 2.7B | base | 7.9 | 6.5 | 11.8 | 20.7 | .2 | .6 | .6 |
| | | +extra | 6.7 | 6.0 | 9.0 | 16.8 | .2 | .6 | .6 |
| PolyCoder [70] | 2.7B | base | 6.1 | 5.9 | 10.2 | 17.1 | .2 | .4 | .6 |
| | | +extra | 5.5 | 5.3 | 7.9 | 13.6 | .2 | .6 | .6 |
| StableLM [60] | 7B | base | 2.4 | 2.7 | 7.5 | 15.8 | .2 | .6 | .6 |
| | | +extra | 2.4 | 2.6 | 6.2 | 11.9 | .2 | .6 | .6 |

Table 4: Reduced test-suite for HUMANEVAL⁺. We first show the pass@1⋆ and average #tests (including base HUMANEVAL tests) by only doing set covering over each considered metric separately (§2.2). The **Full** column then shows the final reduction result by combining all of the three. For reference, the average #tests of original HUMANEVAL and HUMANEVAL⁺ are 9.6 and 774.8 respectively (Table 2).

| | Size | Coverage | | Killed mutants | | Killed samples | | **Full** | | Ref. pass@1⋆ | |
|---|---|---|---|---|---|---|---|---|---|---|---|
| | | pass@1⋆ | #tests | pass@1⋆ | #tests | pass@1⋆ | #tests | pass@1⋆ | #tests | base | +extra |
| GPT-4 | N/A | 86.0 | 11.3 | 82.9 | 11.4 | 78.7 | 13.8 | **78.0** | 16.1 | 88.4 | 76.2 |
| ChatGPT | N/A | 71.3 | 11.3 | 69.5 | 11.4 | 65.2 | 13.7 | **65.2** | 16.0 | 73.2 | 63.4 |
| StarCoder | 15B | 32.9 | 11.3 | 32.9 | 11.4 | 29.3 | 13.6 | **29.3** | 15.9 | 34.1 | 29.3 |
| CodeGen | 2B | 23.2 | 11.3 | 23.8 | 11.4 | 21.3 | 13.2 | **21.3** | 15.4 | 24.4 | 20.7 |
| | 6B | 28.7 | 11.3 | 29.3 | 11.4 | 25.6 | 13.2 | **25.6** | 15.4 | 29.3 | 25.6 |
| | 16B | 31.7 | 11.3 | 31.1 | 11.4 | 27.4 | 13.2 | **27.4** | 15.4 | 32.9 | 26.8 |
| CodeGen2 | 1B | 10.4 | 11.3 | 11.0 | 11.4 | 9.1 | 13.8 | **9.1** | 16.0 | 11.0 | 9.1 |
| | 3B | 15.9 | 11.3 | 15.9 | 11.4 | 12.8 | 13.8 | **12.8** | 16.0 | 15.9 | 12.8 |
| | 7B | 18.3 | 11.3 | 18.3 | 11.4 | 16.5 | 13.8 | **16.5** | 16.0 | 18.3 | 16.5 |
| | 16B | 19.5 | 11.3 | 18.9 | 11.4 | 16.5 | 13.8 | **16.5** | 16.0 | 19.5 | 16.5 |
| VICUNA | 7B | 11.6 | 11.3 | 11.6 | 11.4 | 11.0 | 13.8 | **11.0** | 16.1 | 11.6 | 10.4 |
| | 13B | 16.5 | 11.3 | 16.5 | 11.4 | 15.2 | 13.8 | **15.2** | 16.1 | 17.1 | 15.2 |
| SantaCoder | 1.1B | 14.6 | 11.3 | 14.6 | 11.4 | 12.8 | 13.8 | **12.8** | 16.1 | 14.6 | 12.8 |
| INCODER | 1.3B | 12.2 | 11.3 | 12.2 | 11.4 | 10.4 | 13.6 | **10.4** | 16.0 | 12.2 | 10.4 |
| | 6.7B | 14.6 | 11.3 | 14.6 | 11.4 | 12.2 | 13.6 | **12.2** | 16.0 | 15.9 | 12.2 |
| GPT-J | 6B | 12.2 | 11.3 | 12.2 | 11.4 | 10.4 | 13.8 | **10.4** | 16.0 | 12.2 | 10.4 |
| GPT-NEO | 2.7B | 7.3 | 11.3 | 7.3 | 11.4 | 6.7 | 13.8 | **6.7** | 16.1 | 7.9 | 6.7 |
| PolyCoder | 2.7B | 6.1 | 11.3 | 6.1 | 11.4 | 5.5 | 13.8 | **5.5** | 16.1 | 6.1 | 5.5 |
| StableLM | 7B | **2.4** | 11.3 | **2.4** | 11.4 | **2.4** | 13.8 | **2.4** | 16.1 | 2.4 | 2.4 |

We also show that a more rigorous evaluation could yield different or totally contradictory relative results. For example, WizardCoder-CodeLlama and Phind-CodeLlama on the original HUMANEVAL are evaluated to be no better than ChatGPT in terms of pass@1⋆. However, HUMANEVAL⁺ demonstrates that the two open-source models can actually outperform the proprietary ChatGPT. Other contrary examples reflected by HUMANEVAL⁺ include that SantaCoder-1B surpasses INCODER-6.7B and VICUNA-7B outperforms INCODER-1.3B. Table 3 further illustrates the distribution of best-performing temperatures over different $k$ values. Our results conforms with prior findings [11] that a lower temperature tends to perform better for smaller $k$, while a higher temperature works better for larger $k$. We also observe that the optimal temperatures seem to stay fairly consistent before and after using HUMANEVAL⁺; however, slight differences still exist, *e.g.,* best temperature for CodeGen-2B on pass@10 becomes 0.2 from 0.8 after using HUMANEVAL⁺. Nonetheless, this motivates future research to look more closely on the effect of temperature with respect to the robustness of the evaluation tests, esp. those edge-cases.

**Effectiveness of test-suite reduction.** Based on HUMANEVAL⁺ which on average obtains 764.1 tests for each programming task (Table 2), our test-suite reducer (§2.2) minimizes it to HUMANEVAL⁺-MINI which only has 16.1 tests for each task (smaller by $47\times$). Table 4 performs leave-one-out cross validation to show the pass@1⋆ differences over a subset of representative models studied in Table 3 (due to time/space constraints). That is, for each evaluated LLM we construct the reduced test-suite without considering its own sample kills. The **Full** column shows that the reduced test-suite can achieve almost the same pass@1⋆ drop as HUMANEVAL⁺ by only using $47\times$ fewer test-cases. Taking a closer look, separately performing set covering over each metric can harness the pass@1⋆ of the base HUMANEVAL to certain degree. Specifically, the use of empirical LLM sample killings is the most effective, leading to the same effectiveness as the full approach, but also consumes more tests than other theoretical metrics. While using coverage and mutation analysis seems to be unnecessary in addition to using sample killings, they still serve as the base guarantees for the theoretical test adequacy.

**Pass rate distribution.** Figure 3 shows for each programming task the overall pass rates on HUMANEVAL and HUMANEVAL⁺ tests. The pass rate gap between HUMANEVAL and HUMANEVAL⁺ shows overall HUMANEVAL⁺ can detect solutions that are misidentified by HUMANEVAL for problems of all levels of difficulties. We also observe that problems in HUMANEVAL are not equal, not only in terms of problem difficulty but also the difficulty of generating counter-examples and

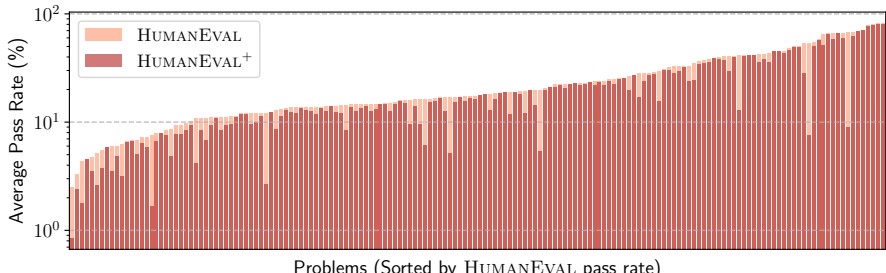

Figure 3: Pass rate distribution. X-axis spans bars for all 164 problems, sorted by the HUMANEVAL pass rate. Y-axis shows the *log*-scale pass rates averaged by all LLM-generated samples.

edge-cases to deeply exercise LLM-generated code. For simple problems such as "adding two numbers" and "length of a string" (*i.e.,* problems with top-2 pass rates), it is easy to solve for LLMs and to test manually. While problems dealing with *multiple conditions* (*e.g.,* "word splitting"), *completeness* (*e.g.,* handling negative numbers for "is-prime") , *reasoning ability* (*e.g.,* "Tribonacci sequence") and *efficiency requirements* (*e.g.,* "n-th prime Fibonacci number") are the hardest tasks to the evaluated LLMs, positioning future research to improve LLMs for conquering such coding skills.

**Incorrect "ground-truth" in HUMANEVAL.** In addition to detecting wrong code from LLMs using EvalPlus, we also found **18** defects (11% of problems) even in the original ground-truth in HU-MANEVAL, including *(i) Unhandled edge-case*: five prior ground-truths fail to handle corner-case inputs (*e.g.,* empty list or string); *(ii) Bad logic*: 10 prior ground-truths incorrectly implement the desired functionality; and *(iii) Performance issue*: three inefficient implementations lead to slow performance on reasonably-sized inputs. Among those, bad logic (10) is the most serious as the original "ground-truth" does not accurately reflect the user intent. Such defects are detected also through differential testing but between our own re-implemented ground-truth and the original ground-truth in HUMANEVAL.

```
def valid_date(date):
    ...
    if month in [1,3,5,7,8,10,12] and day < 1 or day > 31:
        return False
    if month in [4,6,9,11] and day < 1 or day > 30:
        return False
    ...
```

12-31-1999 → → False

HUMANEVAL⁺ input     A bracket is needed!     12/31/1999 is a valid date!

Figure 4: Exemplary incorrect-logic ground-truth solution in HUMANEVAL (#124)

Figure 4 shows an incorrect ground-truth implementation (`validate_date`) from HUMANEVAL classified as having bad logic. The desired task is to check if the input date format is correct. We see that in the core logic, the conditions attempt to first check the `month` condition and then handle the corresponding `day` conditions. However, this is implemented incorrectly as "`and`" in Python[5] has higher precedence than "`or`", leading to the ground-truth function to check if *either* conditions satisfies instead of the desired *both* conditions must satisfy. This is exposed via our automatically generated test input of `12-31-1999` where the ground-truth implementation incorrectly labels this as not a valid date. Surprisingly this egregious error is not exposed by any of the base test inputs in HUMANEVAL, further demonstrating the weakness and limited evaluation power of the original test inputs.

## 4   Related Work

**LLMs for code**. The use of LLMs for code has gained traction in recent years, owing to the abundance of open codebase and the need for improving developer efficiency. LLMs have demonstrated state-of-the-art performance on various code-related tasks, including code generation [11, 33, 25], program repair [69, 27, 68, 65], automated testing [15, 14, 67, 35, 71], code translation [31, 55] and code summarization [1, 37]. In particular, prominent LLMs including CODEX [11], CodeGen [46], INCODER [18] and PolyCoder [70], have been developed and extensively evaluated for code

[5]https://docs.python.org/3/reference/expressions.html#operator-precedence

generation (widely recognized as the holy grail for computer science research since the inception of AI in the 1950s [21]), where the model generates code snippets based on natural language descriptions (*e.g.,* docstring) of the desired functionality.

**Coding benchmark for LLMs.** LLM-based code synthesis is largely evaluated based on functional correctness, which is typically assessed by running test-cases to check the desired outputs. HUMANEVAL [11] is one of the pioneering and most widely studied human-written benchmarks for LLM-based code synthesis, consisting of 164 pairs of Python function signature with docstring and the associated test-cases for correctness checking. Additionally, each HUMANEVAL problem is also equipped with a reference solution. Another Python-focused dataset, MBPP [3], is created by crowd-sourcing participants to write in summation 974 programming problems, each of which is comprised of the problem statement (*i.e.,* docstring), the function signature, as well as *three* test-cases. Beyond Python, there are other benchmarks targeting additional languages such as Spider [73] (SQL), HUMANEVAL-X [76] (C++, Javascript and Go), CodeContests [33] (C++ and Java) and MultiPL-E [9] (extending HUMANEVAL and MBPP to 18 programming languages). More recently, researchers have created a more realistic code synthesis benchmark by collecting GitHub issues along with the corresponding code base together with tests to measure the ability of LLMs to perform real-world software engineering tasks [28]. Our work shows for the first time the test inadequacy problem of widely studied benchmarks and addresses the issue via automatic test generation.

**Automated test generation**. Automated test generation is a widely used for finding software bugs with automatically generated tests. *Black-box* test generation such as fuzz testing [43] feeds random inputs (*e.g.,* random bytes) to the system under test (SUT), without knowing its source code. Traditional black-box techniques can mainly be categorized into generation-based [72, 23, 56] and mutation-based [66, 10, 47] ones. *White-box* approaches provide better-quality test-cases by analyzing the source code of SUT. For instance, symbolic execution [30, 8] breaks the coverage plateaus by solving symbolic path constraints to generate tests targeting deep paths. As a mid-point, coverage-guided fuzzing [74, 57] (*i.e.,* grey-box) uses the coverage information of SUT as feedback to adjust the input generation and mutation. The discussed traditional methods are inapplicable to generating semantically meaningful inputs for arbitrary problems programmed in a dynamically-typed language. We address this by using ChatGPT to inspect the ground-truth (*i.e.,* white-box) for initializing interesting seeds, based on which type-aware mutation (*i.e.,* black-box) scales the test inputs to a large amount.

# 5 Conclusion & Future Work

We present EvalPlus – a rigorous evaluation framework for program synthesis, driven by automated test generation. EvalPlus combines both LLM- and mutation-based input generation to obtain a diverse set of test inputs for accurately evaluating the correctness of LLM-generated code. EvalPlus creates HUMANEVAL⁺, built on top of the popular HUMANEVAL with additional high-quality and automatically generated test inputs. With test-suite reduction, EvalPlus also produces HUMANEVAL⁺-MINI which is smaller than HUMANEVAL⁺ by $47\times$ while preserving similar test effectiveness. We extensively evaluate a diverse set of LLMs and show that HUMANEVAL⁺ can identify a significant amount of previously undetected wrong code generated by LLMs, demonstrating its effectiveness to augment programming benchmarks for more accurate evaluation.

Since launched, the EvalPlus PyPI package has been installed by over 6k times in 5 months. We also keep evaluating new models for code and maintain a leaderboard at [https://evalplus.github.io/leaderboard.html](https://evalplus.github.io/leaderboard.html). In the future, we plan to apply EvalPlus to bring better-quality testing for more code benchmarks such as MBPP. Meanwhile. future work can look into how to integrate EvalPlus with more formal verification (*e.g.,* Dafny [32]) or validation techniques (*e.g.,* translation validation [36]) to provide stronger guarantees of the evaluation results when applicable. Additionally, the core test generation technique behind can be even used to remind developers of potential flaws of the accepted LLM-generated code snippets when doing AI pair-programming (*e.g.,* Copilot [42]).

# 6 Acknowledgements

This work was partially supported by NSF grants CCF-2131943 and CCF-2141474, as well as Kwai Inc. We thank the reviewers for their invaluable feedback. We further thank Yinlin Deng for providing helpful discussions, as well as Junhao Wang and Songrun Xie for their open-source contributions.

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

Table 5: Overview of evaluated models.

| | Model Name | Sizes | Release Year | Open-Source |
|---|---|---|---|---|
| Coding | CodeGen [46] | 2B, 6B, 16B | 2022 | ✓ |
| | INCODER [18] | 1.3B, 6.7B | 2022 | ✓ |
| | PolyCoder [70] | 2.7B | 2022 | ✓ |
| | SantaCoder [2] | 1.1B | 2023 | ✓ |
| | CodeGen2 [45] | 1B, 3B, 7B, 16B | 2023 | ✓ |
| | StarCoder [13] | 15B | 2023 | ✓ |
| | CODET5+ [64] | 16B | 2023 | ✓ |
| | CODELLAMA [54] | 7B,13B,34B | 2023 | ✓ |
| | WizardCoder-CodeLlama [38] | 34B | 2023 | ✓ |
| | Phind-CodeLlama [52] | 34B | 2023 | ✓ |
| General | GPT-J [63] | 6B | 2021 | ✓ |
| | GPT-NEO [5] | 2.7B | 2021 | ✓ |
| | ChatGPT [48] | N/A | 2022 | |
| | GPT-4 [49] | N/A | 2023 | |
| | VICUNA [12] | 7B, 13B | 2023 | ✓ |
| | StableLM [60] | 7B | 2023 | ✓ |
| | MISTRAL [26] | 7B | 2023 | ✓ |

# A  Detailed Experimental Setup

**Evaluation of LLMs.** Our goal is to comprehensively evaluate recent and widely used LLMs, both specialized for code generation [46, 70, 18, 2, 52, 38, 64] and general-purpose tasks [49, 48, 12, 60, 63, 5, 26]. Table 5 presents an overview of the studied models, with column **Sizes** reflecting the model sizes in billions of parameters, **Release Year** showing when the LLM is released, and **Open-Source** marking the models whose weights are publicly available. In total, we evaluate 26 of the most representative and popular LLMs with a broad range of configurations to fully demonstrate the generalizability of our results.

Our hyper-parameter configurations follow prior work [11, 46]. For each model we randomly sample 200 programs and repeat the experiments over temperature ($\{0.2, 0.4, 0.6, 0.8\}$) and greedy decoding with zero temperature. By default, we let each model generate at most 512 new tokens and truncate the produced code with end-of-string (EOS) identifiers suggested in HUMANEVAL [11], as well as those favoured by certain models (*e.g.,* "`<|endoftext|>`" and "`\n```"). For conversational models (*i.e.,* ChatGPT and GPT-4), we obtain the code fragments by parsing the code blocks (*i.e.,* within "``` ") in the output. We found ChatGPT tends to repeat problem description with detailed explanation, which can consume more than 512 new tokens to complete a solution for around 11% of problems. To align ChatGPT with other models, for tasks with very long problem descriptions, we extend the token limit from 512 to 1024. For model implementation, we run ChatGPT and GPT-4 via OpenAI APIs, and accelerate CodeGen-6B and -16B with NVIDIA FasterTransformer via FauxPilot [16]. All other LLMs are based on the HuggingFace `transformers` library. By default, we follow the official examples of each LLM (*e.g.,* on HuggingFace model card) to construct their corresponding prompts. Specifically, the prompts used for ChatGPT, GPT-4, and WizardCoder-CodeLlama is instruction-based, *i.e.,* a simple instruction is used to wrap the function signature and docstring to explicitly encourage the LLM for code generation.

**Test oracles.** An LLM-produced solution is regarded to be correct if for all test inputs it returns values that match the expected outputs within a reasonable run time. We perform exact matching by default. For floating-point comparisons, we tolerate absolute differences to the degrees annotated in HUMANEVAL or $10^{-6}$ if not annotated. In original HUMANEVAL, the default timeout is set to three seconds to run the whole test-suite (*i.e.,* all test-cases) for each programming problem. Such a setting is neither suitable when having more test-cases nor reasonable as each problem could have its own run time characteristics. Consequently, we let the timeout for each test-case to be $\max(200\text{ms}, 4 \times t_{gt})$ where $t_{gt}$ refers to the execution time of the corresponding ground-truth solution. In other words, we expect the LLM-provided solution to be no slower than the ground-truth by four times or use a base 200-millisecond timeout when $4 \times t_{gt} < 200\text{ms}$ to avoid variance caused by performance randomness.

