# OpenReview forum: "Is Your Code Generated by ChatGPT Really Correct? Rigorous Evaluation of Large Language Models for Code Generation"
_NeurIPS.cc/2023/Conference — NeurIPS 2023 poster_

### Official Review · Reviewer_jsgr · 2023-06-26

**Soundness:** 3 good
**Presentation:** 3 good
**Contribution:** 3 good
**Rating:** 5
**Confidence:** 4

**Summary:**

This paper leveraged large language model (LLM) based and mutation-based strategies to generate high-quality test cases for the popular dataset HumanEval. The extended dataset HumanEval+ provides a better code generation benchmark for assessing the performance of LLMs such as ChatGPT and GPT4. Experimental results showed that compared to the HumanEval dataset, the extended one can detect more issues of the generated code, where the generation performance of 19 LLMs is reduced by 13.6-15.3% on average in terms of pass@k. To save the testing time, this paper also provided a refined set of test cases with fewer numbers but with the same code coverage, killed mutants, and sample killings.



**Strengths:**

+ Proposing a test case generation method for LLMs, which combines the LLM-based and mutation-based strategies.

+ Evaluating 19 LLMs on the extended datasets and showing the overestimation of the original dataset.

+ Providing a test-suite reduction for quick evaluation.


**Weaknesses:**

- Some technical details, such as the quality of seed inputs, are unclear, which affect the soundness of the proposed approach.

- There are some unclear claims/overclaims.

- Lack of comparisons with other test case generation methods and related work.

The claimed contribution indicates “that prior popular code synthesis evaluation results do not accurately reflect the true performance of LLMs for code synthesis”. Can the extended dataset proposed by this paper reflect the “true performance”? Similarly, the paper said that the existing tests “often fall short in capturing all possible scenarios”, leading to “false confidence” in the results. Can your test cases “capturing all possible scenarios”? Furthermore, this paper claimed that the proposed benchmark can “precisely evaluate the functional correctness of LLM-generated code” by generating “interesting test inputs”. It is unclear to me how “precise” can the evaluation provide and what are the “interesting test inputs”. Please clarify them.

EvalPlus “first uses ChatGPT to generate a set of high-quality seed inputs for later mutation”. The quality of the seed inputs is not verified. Are these inputs correct? How do you ensure the quality of the generated seed inputs? and how many seed inputs did the ChatGPT generate? What are the prompts used?

The type-aware input mutation “randomly” selected some inputs from the generated seed pool as the inputs of mutants. This pipeline did not clearly describe how randomness affects the result of the input mutation and the quality of the evaluation.

The proposed approach “adopted a programming by contract philosophy by systematically adding code assertions as contracts (e.g., assert n > 0) to ensure the test inputs for the function are well-formed”. The context did not tell how to define the assertions specifically, how to ensure the correctness of the definitions, and which docstrings need clarification with these assertions. Will it incur a lot of manual effort?

For the evaluation, the paper did not compare the method with other baseline methods. The paper says that traditional automated test generation methods are inapplicable to generating semantically meaningful inputs. Why is that so? More descriptions are needed here. Otherwise, it is difficult to determine the advantages of the proposed method over the traditional methods.

Recently, there is also a related work on applying ChatGPT prompts to code generation, the authors also discussed the quality/correctness of the generated code: C. Liu et al., Improving ChatGPT Prompt for Code Generation, https://arxiv.org/abs/2305.08360.


**Questions:**

How is the quality of the seed inputs generated by ChatGPT? Are these inputs correct? How do you ensure the quality of the generated seed inputs? and how many seed inputs did the ChatGPT generate? What are the prompts used? Will the quality of seed inputs affect your approach significantly?

**Limitations:**

 The authors addressed the limitations in the Appendix.

---

> ### Author Rebuttal · Authors · 2023-08-09
>
> > Q1:How is the quality of the seed inputs generated by ChatGPT and will the quality of seed inputs affect your approach significantly?
>
> The quality of seed inputs and the effect it has on mutation-based test generation and fuzzing has been well-studied in prior work [5, 6]. Similarly, the quality of seed inputs will also affect our approach significantly, and that is exactly why we are leveraging ChatGPT to generate high-quality seed inputs. We found that compared to the original HumanEval test, by adding the 30 high-quality seeds generated by ChatGPT, we can already improve coverage from 96.7% to 98.4%, and finally decrease the average pass@k by 11% (full EvalPlus is around 14%). This demonstrates the quality of the seed inputs generated by ChatGPT.
>
> [5] Rebert, Alexandre, Sang Kil Cha, Thanassis Avgerinos, Jonathan Foote, David Warren, Gustavo Grieco, and David Brumley. "Optimizing seed selection for fuzzing."
>
> [6] Pailoor, Shankara, Andrew Aday, and Suman Jana. "MoonShine: Optimizing OS fuzzer seed selection with trace distillation."
>
> > Q2:Are these inputs correct?
>
> The raw inputs generated by ChatGPT are not guaranteed to be correct. However, as mentioned in Section 2.3, EvalPlus uses program input contracts to filter out ill-formed inputs and the ones satisfying contract conditions are bound to be correct.
>
> > Q3:how many seed inputs did the ChatGPT generate?
>
> Please kindly see Section 3 for the discussion. For each programming task, we generate 30 seed inputs using ChatGPT.
>
> > Q4:What are the prompts used?
>
> We apologize for not including the detailed prompt in our supplementary material (we did show an overview of the prompt in Figure 2), and will include the detailed prompt in the appendix of the next revision. Below is our detailed prompt used for seed generation.
> ```
> Here is a function that we want to test:
> [[FUNCTION]]
> These are some example inputs used to test the function:
> [[EXAMPLES]]
> [[INSTRUCTION]]
> ```
> Specifically, the meanings of these macros are:
> [[FUNCTION]]: The ground-truth solution of the programming task
> [[EXAMPLES]]: (At most) 5 inputs randomly selected from original HumanEval
> [[INSTRUCTION]]: We randomly select one of the following instructions:
> “Please generate complex inputs to test the function”
> “Please generate corner case inputs to test the function”
> “Please generate difficult inputs to test the function.”
>
> > C1: Lack of comparisons with other test case generation methods and related work.
>
> We first want to clarify that our main contribution is not our test generation techniques but instead is our generated dataset and accompanying rigorous evaluation study on recent popular LLMs, which demonstrates for the first time that prior popular benchmarks are insufficient to evaluate functional correctness of LLM generated programs. Regarding the comparison with prior test case generation methods, there has been a ton of work on automatically generating tests for programs [7, 8] (see Section 4 for more detail). These prior work mainly focused on either generating new inputs (generation-based) or mutating existing seeds (mutation-based) to create new test cases. Unfortunately, discussed traditional methods are mostly inapplicable to generating semantically meaningful inputs for arbitrary problems programmed in a dynamically-typed language. We address this by using ChatGPT to inspect the ground-truth (i.e., white-box) for initializing interesting seeds, based on which type-aware mutation (i.e., black-box) scales the test inputs to a large amount. Of course, we fully agree with the reviewer that our work may inspire more test generation techniques for this important domain. We will also work towards adding more discussion and comparison with previous test case generation methods in the next revision.
>
> [7] Fraser, Gordon, and Andrea Arcuri. "Evosuite: automatic test suite generation for object-oriented software."
>
> [8] Serebryany, Kosta. "Continuous fuzzing with libfuzzer and addresssanitizer."

---

> > ### Comment · Reviewer_jsgr · 2023-08-22
> >
> > Thanks for the rebuttal. I will keep my current score.

---

> > > ### Author Response · Authors · 2023-08-22
> > >
> > > Thanks for taking the time to read our response thoroughly! We truly appreciate it and will address all minor comments raised by the reviewer.

---

> ### Comment · Area_Chair_GnYf · 2023-08-18
>
> Reviewer, please confirm that you read this rebuttal and adjusted your score and review if appropriate.

---

### Official Review · Reviewer_VxVU · 2023-07-07

**Soundness:** 3 good
**Presentation:** 3 good
**Contribution:** 3 good
**Rating:** 6
**Confidence:** 4

**Summary:**

In this paper, the authors introduce EvalPlus, an evaluation framework crafted to assess the code generation of LLMs. Combining LLM and mutation-based techniques, EvalPlus diversifies the generated test inputs, thereby broadening the evaluation spectrum for LLM-produced code. Through comprehensive experimentation, the authors highlight EvalPlus's ability to expose shortcomings in previous LLM-based code evaluations. Additionally, the study brings HumanEval Plus, an enriched dataset built upon the existing HumanEval, offering an abundance of test inputs and a more dependable ground truth for improved reliability.

**Strengths:**

* This paper distinguishes itself through its innovative approach to test input generation. It extends beyond the conventional usage of ChatGPT for creating test inputs, by incorporating a test-mutation technique and a “distill” method. This results in a diverse yet non-redundant range of test inputs, illustrating the authors' commitment to crafting a robust and efficient evaluation framework.
* This paper is well-written and easy to read.
* The study stands out for its well-designed experiment, which encompasses a comprehensive selection of contemporary LLMs.

**Weaknesses:**

* HumanEval [8], in comparison to real-world coding projects, presents a significantly simpler challenge. It's a benchmark that's even simpler than others, such as APPS [Hendrycks et al.]. Given this context, the effectiveness of EvalPlus when applied to more complex tasks remains uncertain. This indicates a valuable area for enhancement in this study—specifically, future research could delve into assessing how well EvalPlus performs in more complicated, real-world coding scenarios. Such explorations could contribute crucial insights to understanding the scalability and adaptability of EvalPlus.

**Questions:**

* Given the impressive results demonstrated by EvalPlus within the comparatively simpler context of the HumanEval benchmark, how do you envision expanding this work to more complex environments? Specifically, what strategies or modifications do you plan to implement to ensure EvalPlus is effective in assessing code generated for real-world projects, such as those found in open-source Python libraries? Could you elaborate on potential challenges you anticipate and how you intend to navigate them in this expansion?

**Limitations:**

* While this study presents a promising step forward with EvalPlus, a limitation lies in the evaluation scenario chosen for testing. Given that the proposed method is intended to be general, using the relatively simple HumanEval dataset for testing may not sufficiently demonstrate the framework's generalization capabilities. To conclusively assert its wide applicability and robustness, it would be beneficial for future research to include tests on more complex datasets or in more challenging real-world contexts, such as open-source Python libraries. This would provide a more comprehensive understanding of EvalPlus's potential in diverse, practical applications.

---

> ### Author Rebuttal · Authors · 2023-08-09
>
> >  Q1:how do you envision expanding this work to more complex environments? (strategies, modifications and challenges to ensure EvalPlus is effective in assessing code generated for real-world projects?)
>
> To begin with, we want to re-emphasise that our main contribution is to show that the existing popular benchmarks for LLM-based code synthesis evaluation (e.g., HumanEval) contain insufficient tests that cannot faithfully evaluate the functional correctness for LLM generated code. As such, the reported performance of LLMs when evaluated on these benchmarks can be inaccurate or exaggerated (compared to their actual performance). For example, this work shows for the first time that almost all recent LLMs for code can be affected by such dataset issues. Furthermore, the paper also helps more precisely understand the strengths/limitations of existing LLMs via automated test generation/reduction, and can in turn help build more powerful LLMs in the near future.
>
> Meanwhile, this is a great question that allows us to discuss potential future work! First, we plan to expand EvalPlus by taking in more knowledge and context from the entire project rather than a single function. For example, developer documentation can be directly used as input to help ChatGPT synthesize interesting inputs that not only invokes a single function but also multiple functions or API sequences. Second, one key challenge is that unlike the benchmarks that we seek to improve in this paper, real-world projects do not have the exact groundtruth solution. In order to address this challenge, we can instead apply partial test oracles such as 1) crashes: to discover rare inputs that trigger crashes in developer code (e.g,. segmentation faults), 2) differential testing: to evaluate developer code on two different setups (i.e. CPU vs GPU for deep learning programs) to discover bugs, and 3) LLM-based oracle generation: we can even leverage the generative and code understanding power of LLMs themselves to generate the oracle.These partial oracles may not give us the same guarantees as using the reference groundtruth solution, but EvalPlus can still leverage these oracles to assist developers in more complex real-world systems.

---

> ### Comment · Area_Chair_GnYf · 2023-08-18
>
> Reviewer, please confirm that you read this rebuttal and adjusted your score and review if appropriate.

---

### Official Review · Reviewer_9v6g · 2023-07-07

**Soundness:** 3 good
**Presentation:** 4 excellent
**Contribution:** 3 good
**Rating:** 6
**Confidence:** 3

**Summary:**

The authors propose an evaluation framework for validating the correctness of large language model-generated code. In particular, the framework first utilizes ChatGPT to generate multiple seed inputs, which are then expanded into a large set of inputs through type-aware mutation. In addition, to ensure evaluation efficiency, the input set can be reduced by employing code coverage, mutant killings, and LLM sample killings.

**Strengths:**

The proposed solution addresses the problem of insufficient testing for LLM-generated code. The expanded dataset can outperformance the original one, while the reduced dataset achieves similar results as the expanded one.
The authors evaluate the proposed solution via comparative analysis of 19 large language models, analyze the pass rate distribution of the employed dataset HUMANEVAL, and identify several errors in the ground-truth solutions.
The related work section discusses previous research on large language models for code, the coding benchmark, and automated test generation. The authors discuss the research gap between this study and the related work.

**Weaknesses:**

The authors do not employ a more capable large language model in the proposed solution, and there is a lack of evaluation of efficiency.

**Questions:**

- I wonder if GPT-4 can generate more interesting seed inputs than ChatGPT. Explicitly clarifying the employed component in EvalPlus may affect the generality of the solution, I suggest this can be introduced as the implementation or setup information.
- The authors can elaborate on how to filter out the invalid inputs in Section 2.1.
- The evaluation does not examine the efficiency of type-aware input mutation, and test-suite reduction.
- Table 4: “Killed samples” has ambiguity.
- Section 2: “EvalPlus obtains a augmented benchmark …” -> “an”.


**Limitations:**

The authors do not explicitly identify the potential societal impact or limitations of this work, but they claim that their future work include conducting testing to more code benchmarks.

---

> ### Author Rebuttal · Authors · 2023-08-09
>
> > Q1:I wonder if GPT-4 can generate more interesting seed inputs than ChatGPT.
>
> Please note that the EvalPlus input generation component does not rely only on ChatGPT but is general and can be implemented using any other foundational models like GPT-4 or LaMDA. We have also thought about using GPT-4 to generate additionally interesting seed inputs. However, due to the high monetary cost of invoking the GPT-4 API (on 164 different HumanEval tasks will cost more than 600 dollars compared to only 60 dollars using ChatGPT), we choose to only use ChatGPT for our work. In fact, ChatGPT has already shown impressive performance generating high quality seed inputs. Thanks again for the suggestion and we will further use GPT-4 for seed generation (which will very likely lead to even better performance as mentioned by the reviewer) in the future.
>
> > Q2:Elaborate on how to filter out the invalid inputs in Section 2.1.
>
> This has been elaborated in Section 2.3. In short, we adopt a programming by contract philosophy by systematically adding code assertions as contracts to ensure the test inputs for the function are well-formed. Invalid inputs during generation will trigger assertion failures and thus be detected and dumped by EvalPlus.
>
> > Q3:The evaluation does not examine the efficiency of type-aware input mutation, and test-suite reduction.
>
> Thanks for the suggestion and we will emphasize the efficiency results in our next revision.
> Previously we did not rigorously study the efficiency of input generation and reduction because these are one-time efforts for each dataset (can be reused to evaluate any current or future code models) and can be done reasonably fast. Specifically, EvalPlus is able to generate on average 1100+ valid tests for each of the 164 problems in half an hour (i.e., ~3000 valid tests/hour). The test reduction for all 164 problems in total can also finish in one hour.
>
> > Q4: Table 4: “Killed samples” has ambiguity.
>
> We apologize for the ambiguous term. “Killed samples'' refers to the number of incorrect LLM generated samples that are “killed” (i.e. failed a test) by a test-suite. Please see Section 2.2 for more detail. We will make the term less ambiguous.

---

> > ### Comment · Reviewer_9v6g · 2023-08-17
> >
> > Thanks to the authors for their rebuttal. I have no further comment at this time.

---

> > > ### Author Response · Authors · 2023-08-18
> > >
> > > Huge thanks for taking the time to read our response thoroughly. We truly appreciate it!

---

### Official Review · Reviewer_SGiT · 2023-07-07

**Soundness:** 3 good
**Presentation:** 3 good
**Contribution:** 3 good
**Rating:** 6
**Confidence:** 4

**Summary:**

This paper introduces a rigorous evaluation framework EvalPlus for program synthesis driven by automated test generation. For automated test generation, this work proposes to combine both LLM-generated tests and mutation-based input generation to largely augment the text inputs for an existing code benchmark of HumanEval. EvalPlus is evaluated with a diverse set of code LLMs and results show that the new augmented benchmark can identify a significant amount of previously undetected wrong code generated by LLMs, leading to a more accurate evaluation. It further introduces a mini-version benchmark reserving almost the same test effectiveness.

**Strengths:**

* The paper studies how to evaluate code LLMs more accurately, which has become an important topic in the era of LLMs as coding is one of the key capabilities of LLMs to showcase. The proposed EvalPlus of combining both LLM-generated tests and mutation-based tests is well motivated and technically sound.
* The paper is well written and easy to follow. The evaluation is very comprehensive (considering most of state-of-the-art LLMs) and provides convincing results to authenticate the effectiveness of the proposed EvalPlus.


**Weaknesses:**

* The biggest weakness of this paper is the lack of discussion and comparison to other related work of AlphaCode and CodeT [1], which share common techniques such as mutation-based input generation and unit tests generated by LLMs. In this regard, the novelty of EvalPlus is relatively limited. Note that in AlphaCode, they have already explored similar mutation-based techniques to largely augment the test cases for program synthesis. Besides, they also explored training a LLM-based test input generation model to generate test cases in a clustering process. For CodeT, they have explored using LLMs to generate test cases, though these generated test cases are used for a different purpose of reranking the generated programs. The authors should discuss and compare with these works to better justify their novelties.

[1] CODET: CODE GENERATION WITH GENERATED TESTS


**Questions:**

* Can you explain more on the difference between EvalPlus and AlphaCode/CodeT?
* I noticed that this work compares with the recently released StarCoder and CodeGen2, but not another SoTA code LLM of CodeT5+ which was released at the same time with better results on HumanEval. Any reasons for not including it?


**Limitations:**

I did not find any discussion on limitations from the paper.

---

> ### Author Rebuttal · Authors · 2023-08-09
>
> > Q1:Can you explain more on the difference between EvalPlus and AlphaCode/CodeT?
>
> Great question. First, we want to clarify that our main contribution is not the input generation technique but rather our generated dataset (HumanEval+) and accompanying rigorous study on recent popular LLMs. In short, we demonstrate for the first time that: 1) existing popular datasets (i.e. HumanEval) used to evaluate almost all recent LLMs for code are not reliable, containing not only insufficient tests but also incorrect groundtruths, 2) such deficiencies in existing datasets can drastically affect the evaluation results of almost all recent LLMs for code, with around 15% decrease in performance when using HumanEval+ compared to base HumanEval. EvalPlus can help researchers more precisely understand the strengths and limitations of existing models, and can in turn help build more powerful models in the near future.
>
> In terms of the comparison with AlphaCode and CodeT, while indeed they both use LLM-based input generation, the key difference is that both AlphaCode and CodeT only use the LLM generated inputs for clustering in order to determine the ranking of samples for evaluation. In fact, none of them leveraged the LLM-generated inputs to filter out incorrect solutions, since they do not have an exact oracle to ensure the generated tests are correct. In contrast, in EvalPlus, we directly leveraged the generated test inputs to perform differential testing across the groundtruth implementation and LLM-generated solutions to detect any potential incorrect solutions. Furthermore, compared with the simple mutation operators used in AlphaCode to augment the evaluation tests (different from their LLM-based test input generation that is only for clustering), our type-aware mutation approach also includes an additional mutation operator to collect data fragments, from previously generated inputs, and reuse them during later mutation. This allows our mutation strategy to generate more structurally aware inputs that are likely to pass the structural constraints of certain tasks (e.g., need to be palindrome or open/close bracket strings).
>
> Thanks again for these great references and we will surely discuss more such related work in our revision.
>
> > Q2:I noticed that this work compares with the recently released StarCoder and CodeGen2, but not another SoTA code LLM of CodeT5+ which was released at the same time with better results on HumanEval. Any reasons for not including it?
>
> Additional models are not included due to the lack of time and space. After submission we have been improving EvalPlus and adding more outstanding models including CodeT5+. Specifically, here are the pass@1 (i.e. greedy) results for the 2B, 6B and 16B CodeT5+ models. Overall, our main findings still hold on CodeT5+. Note that using 512 as the maximum new token size (i.e., the default setting of our paper) can cause OOM on A6000-50G for certain problems. Consequently we dynamically reduce the new-token size by 0.8 until OOM is overcome.
>
> | Model |  | pass@1 |
> | :---        |    :----:   |          ---: |
> | CodeT5+ 2B      | base      | 25.0  |
> |  | +extra      | 22.0 (-12%)  |
> | CodeT5+ 6B     | base      |29.3 |
> |  | +extra      | 23.8 (-19%)  |
> | CodeT5+ 2B      | base      | 31.7  |
> |  | +extra      | 26.2 (-17%)  |

---

> ### Comment · Reviewer_SGiT · 2023-08-18
> **Official comment by reviewer SGiT**
>
> Thanks for the detailed response which sufficiently addresses my concerns. I will increase my rating to 6 for this work.

---

> > ### Author Response · Authors · 2023-08-18
> >
> > Big thanks for taking the time to read our response thoroughly. We truly appreciate it! Should you have any new questions or concerns, please don’t hesitate to let us know.

---

### Official Review · Reviewer_yy3N · 2023-07-07

**Soundness:** 2 fair
**Presentation:** 3 good
**Contribution:** 2 fair
**Rating:** 4
**Confidence:** 4

**Summary:**

This paper describes an enhanced test dataset and test-driven evaluation of code generation by LLMs. The paper compares different LLMs and a stricter metrics using the enhanced evaluation dataset and shows that these LLMs are about 15% less correct than is reported based on earlier test datasets.

**Strengths:**

Given that LLMs appear to achieve a number of programming tasks but often fail to completely get them right. They fail in unexpected ways and places. Improving the evaluation metrics is critical to use of LLMs in product and commercial contexts. This paper improves the test beds by synthetically creating addtional tests through a framework that uses both LLMs and mutation based techniques.
The paper demonstrates that about 13-15% of code generated by typical LLMs that would be qualified as passable according to previous metric is disqualified by this method. Since LLMs are often used for NL->code contexts this problem may arise because of inssufficient tests or because of less specific description in NL of the code to be written. They increase the test suites by 81X and also create a reduced version of it that shrinks it 47x for fast evaluation.

The test suites are enhanced by augmenting the prompts with ground truth tests and generating higher quality seed data. From these seed inputs, type-aware mutations are created to enhance the test data sets.




**Weaknesses:**

While the enhanced data set does call out more problems in the LLM-generated code, it is not clear if it is the best it can do. For instance, just by running a static analyser, a syntax checker or some other engine, the issues in the code generated can be found and fixed. The authors should have at least compared one such approach. There is no comparison of another approach that improves the performance of LLMs in context. One of the ways LLMs are often used for code generation is to generate good enough code and then through human interaction or through other static tools rerank, correct or qualify the generated code.
The enhanved data set is diefinitely useful but not clear how useful in that it flags about 13-15% of the code. Some human evaluation or A/B test to qualtitatively say how good this is would have been more insightful.

**Questions:**

1. Have you tried any A/B experiment or other human evaluation to see how much more useful is this improved test set to the LLM context in a real application?
2. Have you thought about other LLM tasks which have similar challenges.
3. As LLMs are constantly improving likely the value of such an enhanced test set diminishes. Have you thought about studying that?

**Limitations:**

Limitations - LLMs are used in context of human input. So it is not clear how useful this enhanced data is. It is also not clear if the enhanced test set increases coverage in criticial dimensions.
LLMs are constantly improving and in the absence of a good A/B test or application context it is hard to tell the usefullness of the system.

---

> ### Author Rebuttal · Authors · 2023-08-09
>
> > Q1:Have you tried any A/B experiment or other human evaluation ...  in a real application?
>
> In this work, we focus on improving the evaluation of functional correctness that can be deterministically, objectively and automatically measured through testing and verification techniques. Contrastively, A/B testing or other human evaluation [1] can be useful for understanding the overall usefulness of NLP applications whose correctness is hard to evaluate systematically, but can incur high manual efforts. Please note that we did in fact manually examine the LLM generated samples that had passed the original HumanEval tests but failed on our EvalPlus tests and found that such solutions can contain subtle but important logic errors (e.g., Figure 1). However, due to the sheer number of samples generated (>45,000), manually performing A/B testing or human evaluation on all samples would be infeasible.
>
> The HumanEval+ benchmark produced by EvalPlus can better reflect the true performance of LLM code synthesis. Not only do we show the average performance drop is around 15%, but even widely used open-source (e.g., CodeGen) and proprietary state-of-the-art LLMs (e.g., ChatGPT/GPT-4) suffer from significant performance decreases (13% to 19%).  As LLMs become more widely used for program synthesis, it is critical to ensure the functional correctness of LLM generated code, with the first step being developing a rigorous benchmark as we did in this work. Furthermore, the input generation proposed in EvalPlus can also be applied to test real-world projects and we hope EvalPlus can inspire more future work in helping developers more rigorously evaluate code for real-world projects.
>
> [1] Zheng, Lianmin, et al. "Judging LLM-as-a-judge with MT-Bench and Chatbot Arena."
>
> > Q2:Have you thought about other LLM tasks which have similar challenges.
>
> Thanks for this interesting question! Comprehensively evaluating LLMs is definitely a common challenge in many LLM tasks. In this work, we target program synthesis as it is the holy grail of computer science since the 1950s, with many recent new LLMs including program synthesis as a key evaluation metric. We believe the general idea and approach of EvalPlus can also help improve datasets, such as MathQA and GSM8K in mathematical reasoning tasks.
> More generally, it is an open research question to apply automated software testing to improve the evaluation of other LLM tasks. For example, consistency checks [2] can automatically evaluate the self-consistency [3] of LLMs, which is a form of “metamorphic testing” -- leveraging multiple inputs and their metamorphic relationship as test oracle. For instance, we can ask an LLM to evaluate two semantically equivalent chess positions to check if the resulting evaluation is also equivalent.
>
> [2] Fluri, Lukasi, et al. "Evaluating Superhuman Models with Consistency Checks."
>
> [3] Wang, Xuezhi, et al. "Self-Consistency Improves Chain of Thought Reasoning in Language Models."
>
> > Q3:As LLMs are constantly improving likely the value of such an enhanced test set diminishes?
>
> It is true that EvalPlus and any software testing technique is useless when we have perfect LLMs. However, please note that this is the ultimate goal for the community and still requires long-term efforts. In fact, our work also aims to contribute towards such an ambitious goal: EvalPlus helps to better evaluate all code models that are still imperfect. We believe it is extremely important to precisely understand the strengths and limitations of existing code models to make informed decisions in improving them, and hope this initial work can inspire more researchers to join this effort.
> Meanwhile, please also note that while models are getting stronger (e.g., ChatGPT to GPT-4), the performance decrease (i.e., the value of such an enhanced test set) does not necessarily become smaller: according to our experiments, while the pass@1 value of ChatGPT drops by 15.5% when using HumanEval+, it drops by 16.0% for GPT-4.
> > C1:just by running a static analyser, the issues in the code generated can be found and fixed.
>
> Thanks for the suggestion. We agree that static code analyzers can detect compile-time (e.g., syntactical error) and simple runtime errors (e.g., wrong arguments). However, code with syntactical errors would already fail the original HumanEval tests, and thus simple syntax checkers cannot help detect any incorrect solutions missed by the original HumanEval. In addition, more advanced static analyzers can only handle very limited types of errors, and are well known to have high false positive rates. For example, Pyright, a state-of-the-art analyzer, can only detect 1 of the 20 incorrect solutions detected by HumanEval+ for GPT-4, and at the same time reported various false positives. As a result, almost all the popular code generation datasets use testing to validate the generated solutions, and EvalPlus also focuses on enhancing tests for such popular datasets.
>
> > C2:no comparison of another approach that improves the performance of LLMs in context.
>
> Great comment. Please kindly note that our main technique is to rigorously evaluate instead of directly improving LLMs. As a result, we focus on evaluating a large spectrum of LLMs. Given the large number of LLMs studied, we only focus on the default application scenario for each model. Meanwhile, since our main findings hold on all the studied LLMs, our results may very likely further generalize to more LLMs or even LLMs with more advanced in-context learning (e.g., with chain-of-thought or execution feedback). For example, even the recent prompting techniques using execution feedback still rely on the existing tests in the dataset, and may still produce incorrect solutions overfitting to the dataset. Thanks again for the comment, and we can surely add more experiments to further validate this argument.

---

> ### Comment · Area_Chair_GnYf · 2023-08-18
>
> Reviewer, please confirm that you read this rebuttal and adjusted your score and review if appropriate.

---

### Author Rebuttal · Authors · 2023-08-09

We thank all the reviewers for their insightful comments and suggestions to improve the paper! We address the main questions (labeled as Q) and concerns (labeled as C) in the response to individual reviewers below. Furthermore, we will also revise the paper accordingly to address all other minor suggestions and comments.

Please kindly let us know if there is any misunderstanding of the questions, and we are very happy to further communicate with all the reviewers during the reviewer-author discussion period (Aug 10-16).

---

> ### Comment · Area_Chair_GnYf · 2023-08-20
> **Reviewers, please respond**
>
> Reviewers, please respond to the rebuttal. The authors state that the primary contribution is the:
>
> > generated dataset and accompanying rigorous evaluation study on recent popular LLMs, which demonstrates for the first time that prior popular benchmarks are insufficient to evaluate functional correctness of LLM generated programs.
>
> The main question for you is whether you think this is of sufficient interest for publication.

---

### Decision · Program_Chairs · 2023-09-21

**Decision:**

Accept (poster)

**Comment:**

The reviewers found this paper to be a valuable contribution. The rebuttal addressed several concerns.